



# Latitudinal Dependence of the Geomagnetic and Solar Activity Effect on Sporadic-E layer

Qiong Tang[1], Chen Zhou[2], Huixin Liu[3], Yi Liu[2], Jiaqi Zhao[1], Zhibin Yu[1], Zhengyu Zhao[1], and Xueshang Feng[1]

[1]Institute of Space Science and Applied Technology, Harbin Institute of Technology, Shenzhen, 518500, China
[2]Department of Space Physics, Wuhan University, Wuhan, 430072, China
[3]Department of Earth and Planetary Science, Kyushu University, Fukuoka, 819-0395, Japan

*Correspondence to*: Chen Zhou (chenzhou@whu.edu.cn)

**Abstract.** Based on the global COSMIC occultation data, the latitudinal dependence of the geomagnetic and solar activity effect on the sporadic-E (Es) layer is investigated. Statistical results demonstrate that the relationship between Es layer occurrence rate and geomagnetic activity shows no correlation in low geomagnetic latitudes, a negative correlation in middle geomagnetic latitudes, and a positive correlation in high geomagnetic latitudes. The decrease in Es layer occurrence rate during geomagnetic activity in middle geomagnetic latitudes may be due to the descending meteor rate caused by the

atmospheric density change during the geomagnetic storm. While the increase in Es layer occurrence rate in high geomagnetic latitudes is mostly related to the ionospheric electric field change driven by the international magnetic field (IMF) embedded within the solar wind. Solar activity effect on the Es layer also presents latitudinal dependence, with negative correlation in low and middle geomagnetic latitudes and positive correlation in high geomagnetic latitudes. The negative correlation may be owing to the negative correlation between meteor rate and solar activity revealed by many

previous studies. The positive correlation in high geomagnetic latitudes is mostly related to the enhanced IMF during solar maximum.

## 1 Introduction

Sporadic E layer, known as Es layer, is an enhanced ionization patch composed of metallic ions ($Fe^+$, $Mg^+$, $Na^+$, $Ca^+$) that appears in the ionospheric E region, mostly at the height range from 90 to 120 km in the mesosphere and the lower

thermosphere (MLT) region. Compared with the electron density of the background ionospheric E layer, the strong vertical electron density gradient and the associated density structures of the Es layer can cause very/ultra high-frequency radio wave scattering and the loss of signal reception (Ogawa et al., 1989; Yue et al., 2016), and hence significantly affect navigation systems, radio communications, and surveillance, etc. (Davies, 1990; McNamara, 1991).

The formation of the Es layer over middle and low latitudes is generally attributed to the wind shear theory, in which the

horizontal winds provide vertical shear points where the long-lived metallic ions, which are debris of the meteoric deposition





and ablation through collisional and electromagnetic processes (Earle et al., 2000; Xue et al., 2013), converge and form a thin layer of intense metallic ionization (Whitehead, 1989; Mathews, 1998; Haldoupis, 2012; Tang et al., 2021).

Equatorial or low-latitude Es layers are classified into two types, one is the same as the mid-latitude Es layer produced by wind shear, and another one is caused by plasma instabilities associated with the equatorial electrojet (Whitehead, 1970, 1989).


At high latitude, the wind shear mechanism is less effective owing to the large inclination of the geomagnetic field. Thus, ionospheric electric fields, which become strong at high latitudes, have been proposed as the dominant effect on drifting metallic ions (Mathews, 1998; Bristow & Watkins, 1991). Two necessary conditions should be satisfied for this primary mechanism to work, namely: electric field direction and an abundance of metallic ions (Nygren et al., 1984, Bedey et al. 1997).


Es layers have been studied over many decades and previous studies reported conflicting results as regards the effect of geomagnetic and solar activities on the Es layer. Either positive, negative or no correlation has been reported by many researchers (Maksyutin & Sherstyukov, 2005; Pietrella & Bianchi, 2009; Zhang et al., 2015; Zhou et al., 2017). Maksyutin and Sherstyukov (2005) revealed that the response of the critical frequency and occurrence of the Es layer to the solar and geomagnetic activities can be both positive and negative. However, Pietrella and Bianchi (2009) found that the occurrence


rate of the Es layer over Rome did not depend on solar and geomagnetic activities. Abdu et al. (2013; 2014) found that the formation and disruption of Es layers over low latitudes were strongly influenced by prompt penetration electric field (PPEF) during the magnetically disturbed period. In recent years, based on coherent radar and digisonde observations, Moro et al. (2017) found that the blanketing Es layer over the equatorial region formed by the wind-shear mechanism could be disrupted


by enhanced electric fields. Therefore, the relationship between the Es layer and the geomagnetic and solar activities remains an open question.

Recently, the radio occultation (RO) technique has been widely used for Es layer observation and investigation. Based on the GPS radio occultation measurements from CHAMP, GRACE-A, and FORMOSAT-3/COSMIC, Arras et al. (2008) first presented the global map of Es distribution and found that the maximum occurrence of the Es layer appears at the middle


latitude regions between about 10° and 60° geomagnetic latitudes during summer hemispheres. Based on the COSMIC S4 index, Yu et al. (2019) showed the global map of Es layer intensity and found that Es intensity shows a high-spatial-resolution geographical distribution and strong seasonal dependence. Thus, the global navigation satellite system is a powerful method for studying the global structure of the Es layer.

Although, Es layer has been extensively investigated, the relation between the Es layer and geomagnetic and solar activities


is still unclear. In the present study, based on the Constellation Observing System for Meteorology, Ionosphere and Climate (COSMIC) occultation data, we aim to study the global Es layer response to the geomagnetic and solar activities, and aim to clear specify the zones of positive, negative and no correlation.



## 2 Datasets

The Constellation Observing System for Meteorology Ionosphere and Climate (COSMIC)/Formosa Satellite 3 (FORMOSAT-3) is a radio occultation mission which is composed of six low earth orbiter micro-satellites. These micro-satellites could provide about 2,000 globally distributed occultation observation profiles per day. The global positioning system (GPS) radio signals are received by the COSMIC precise orbit determination antennas for each GPS-RO when a GPS sets or rises behind Earth's atmosphere, as shown by the LEO satellite. Once the GPS signal is received at the LEO satellite,

the onboard algorithm of the GPS receiver measures SNR intensity fluctuations from the raw 50 Hz L1 amplitude measurements, which are then recorded in the data stream at a 1 Hz rate at the ground receiver to minimize the data record size (Syndergaard et al., 2006). The raw scintillation observations from the receiver are therefore the root mean square of the SNR intensity fluctuation in 1 s (i.e., $\sigma_I$), which can be expressed as $\sigma_I = \sqrt{<(I-<I>)^2>}$. I represents the square of the L1 SNR, and the bracket $<>$ denotes the 1 s average value. The S4 index is reconstructed by the COSMIC Data Analysis and

Archive Center ground processing after these $\sigma_I$ data are downloaded. More details of the procedure of deriving the S4 index can be found in Brahmanandam et al. (2012).

The COSMIC S4 index is defined as the standard deviation of the normalized signal intensity (Briggs & Parkin, 1963). The COSMIC global S4 data include the S4 value and geographic latitude, longitude, altitude, and local time at which S4 is measured. A profile with a maximum value of the S4 (S4max) larger than 0.3 at the 90-130 km altitude range is related to

the occurrence of the Es layer (Yue et al., 2015, 2016; Yu et al., 2019). In this work, the COSMIC-GPS S4 indices during the period from 2007 to 2018 are used to investigate the geomagnetic and solar activity effects on the global Es layer. Solar flux intensity $F_{10.7}$ and geomagnetic index Kp are used for solar and geomagnetic activity level description.

## 3 Result and Analysis

### 3.1 Seasonal and Local Time Variabilities

In Based on the COSMIC-GPS S4 indices from 2007 to 2018 year, the seasonal and local time dependence of Es layer occurrence rate at different geomagnetic latitudinal ranges is investigated first. To evaluate the dependence of the Es layer occurrence rate on seasons, the COSMIC data are categorized into spring (March, April, and May), summer (June, July, and August), autumn (September, October, and November), and winter (December, January, and February). Figure 1 presents the local time-height distribution of Es layer occurrence rate in different seasons in low, middle, and high magnetic latitudes in

both hemispheres. The occurrence rate in each bin is defined as the ratio of the number of Es layer data to the total number of samples. In each subplot of Figure 1, the occurrence rate is binned as a function of local time (LT) from 0 LT to 24 LT with the interval of 1 hour and the height from 90 km to 130 km in the step of 2 km. As shown in Figure 1, the summer maximum of the Es layer occurrence rate is prominent. In addition, the seasonal variation of the Es layer occurrence in middle and high geomagnetic latitudes is more significant than that in low magnetic latitudes since the Es layer occurrence



**Figure 1: The seasonal dependency of Es layer occurrence rate at low, middle, and low magnetic latitudes.**





rate in summer in middle and high magnetic latitudes is far more than in other seasons. As shown by the black curves, the height-local time distribution of Es layer occurrence rate in low geomagnetic latitude range presents helicoid distribution, while that in the middle geomagnetic latitude range shows half spiral structure in the dayside and the nightside, which indicates that the diurnal tidal modulation on Es layer dominants in low latitudes, while the semidiurnal tidal modulation dominants in middle latitudes. Note that the half spiral structure can be observed in the high magnetic latitudes during summer, highlight by the black curves, which indicates that the tidal wind still plays an important role in the formation of the Es layer in high magnetic latitudes. Another important feature is that the Es layer in high magnetic latitudes mainly occurs in the nightside, which is different from the Es layer in middle and low magnetic latitudes that mainly appears in the dayside.

## 3.2 Geomagnetic Variability

Figure 2 shows the geomagnetic activity dependence of the Es layer occurrence in low, middle, and high geomagnetic latitude ranges in both hemispheres. In Figure 2, the data are categorized into three sets of $0 \leq Kp < 3$, $3 \leq Kp < 6$, and $6 \leq Kp < 9$ which represent geomagnetic quiet, geomagnetic moderate, and geomagnetic active conditions. Each panel displays the Es layer occurrence (color-coded) as a function of the local time and altitude of the Es layer from 90 to 130 km in steps of 2 km. We can see from Figure 2 that it is difficult to recognize any increased or decreased trend of Es layer occurrence rate with the increase of Kp index in the low geomagnetic latitude region in both hemispheres. In the middle geomagnetic latitude region of both hemispheres, the Es layer occurrence rate decreases with the enhancement of the geomagnetic activity. However, the Es layer occurrence rate shows an increasing trend with the enhancement of the geomagnetic activity in high magnetic latitude regions.

In order to further study the correlation between the Kp index and Es layer occurrence, we present the variation of the Es layer occurrence rate with the Kp index, as shown in Figure 3. It can be seen from Figure 3 that in the geomagnetic latitude range from 20° S to 30° N, the Es layer occurrence rate shows a zero correlation with the Kp index. In the geomagnetic latitude range between 30° N to 60° N and between 20° S to 60° S, the Es layer occurrence rate shows a negative correlation with the Kp index. However, the Es layer occurrence rate in the geomagnetic latitude region from 60° N to 80° N and from 60° S to 80° S presents a positive correlation with the Kp index. These results indicate that the effect of the geomagnetic activity on the Es layer shows significant latitudinal dependence.

To find out the exact region of zero correlation, negative, and positive correlation, the distribution of the Es layer occurrence rate in the magnetic latitudinal-Kp cross-section is shown in Figure 4. We can see from Figure 4 that in the low geomagnetic latitude region from 25° S to 35° N, the Es layer occurrence rate shows a zero correlation with the Kp index. In the middle geomagnetic latitude ranges from 25° S to 60° S and from 35° N to 60° N, the Es layer occurrence rate decreases with the increase of the Kp index. However, the Es layer occurrence rate presents an increasing trend in the high geomagnetic regions between 60° S and 80° S and between 60° N to 80° N.





**Figure 2: The geomagnetic activity dependency of Es layer occurrence rate at low, middle, and low magnetic latitudes.**





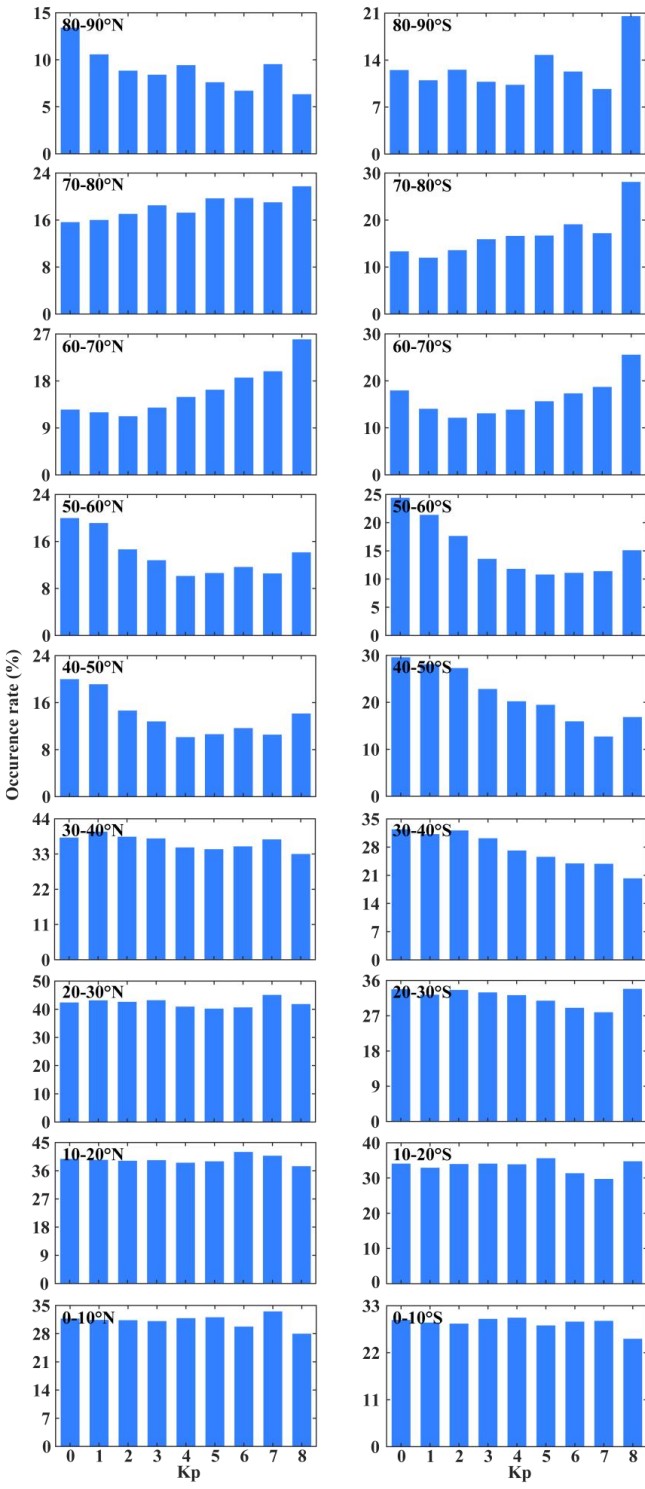


**Figure 3: The variations of Es layer occurrence rate regarding the Kp index at different magnetic latitudinal regions.**





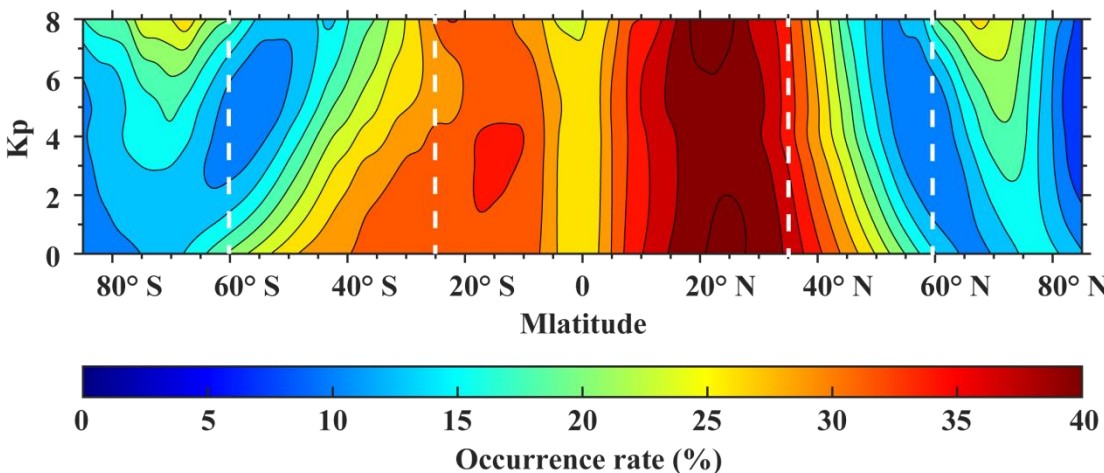

**Figure 4: The distribution of Es layer occurrence rate in magnetic latitudinal-Kp cross-section.**

### 3.3 Solar Variability

Figure 5 shows the solar activity dependence of the Es layer occurrence in low, middle, and high geomagnetic latitude ranges in both hemispheres. The solar activity is divided into three different levels in this study according to the solar flux intensity $F_{10.7}$, including low solar condition ($F_{10.7} \leq 80$), moderate solar condition ($80 < F_{10.7} \leq 110$), and high solar condition ($F_{10.7} > 110$) (Terra et al., 2020). In order to exclude the geomagnetic activity effect, we only analyze the data during the geomagnetic quiet period ($0 \leq Kp < 3$).

As shown in Figure 5, the Es layer occurrence rate in the low geomagnetic latitude regions in both hemispheres presents a decreased trend with the enhancement of solar activity. In addition, the Es layer occurrence rate in the middle geomagnetic latitude regions in the northern hemisphere also decreases with the enhancement of solar activity.

   In order to further study the correlation between the $F_{10.7}$ index and Es layer occurrence, we present the variation of the Es layer occurrence rate with the $F_{10.7}$ index, as shown in Figure 6. In the geomagnetic latitude region from 0° to 60° N and from 0° to 30° S, the Es occurrence rate decreases with the increase of the $F_{10.7}$ index. In the geomagnetic latitude range

above 60° N and 60° S, the Es occurrence rate shows an increasing trend with the enhancement of the $F_{10.7}$ index. Note that in the middle geomagnetic latitude region between 30° S to 60° S, the Es occurrence rate first decreases and then increases with the increase of the $F_{10.7}$ index. These results indicate that solar activity effect on the Es layer also shows latitudinal dependence.

Figure 7 presents the distribution of the Es layer occurrence rate in the geomagnetic latitudinal-$F_{10.7}$ cross-section. In the geomagnetic latitude region from 15° S to 10° N, the Es layer occurrence rate shows a decreasing trend with the increase of the $F_{10.7}$ index. In magnetic latitude ranges above 60° S and 60° N, it shows an increasing trend. Note that the Es occurrence rate in the regions from 15° S to 60° S and from 10° N to 60° N first decreases and then increases with the increase of the $F_{10.7}$ index.



Figure 5: The solar activity dependency of Es layer occurrence rate at low, middle, and low magnetic latitudes.






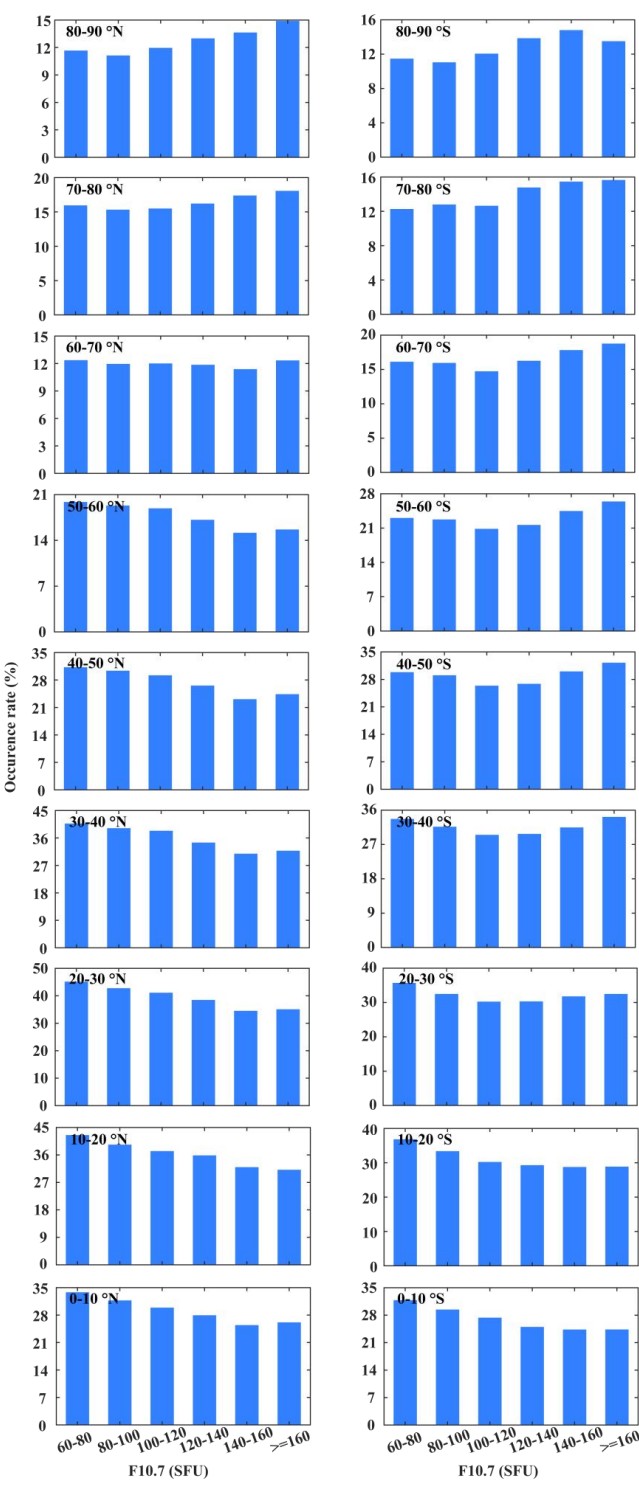

**Figure 6: The variations of Es layer occurrence rate regarding the $F_{10.7}$ index at different magnetic latitudinal regions.**


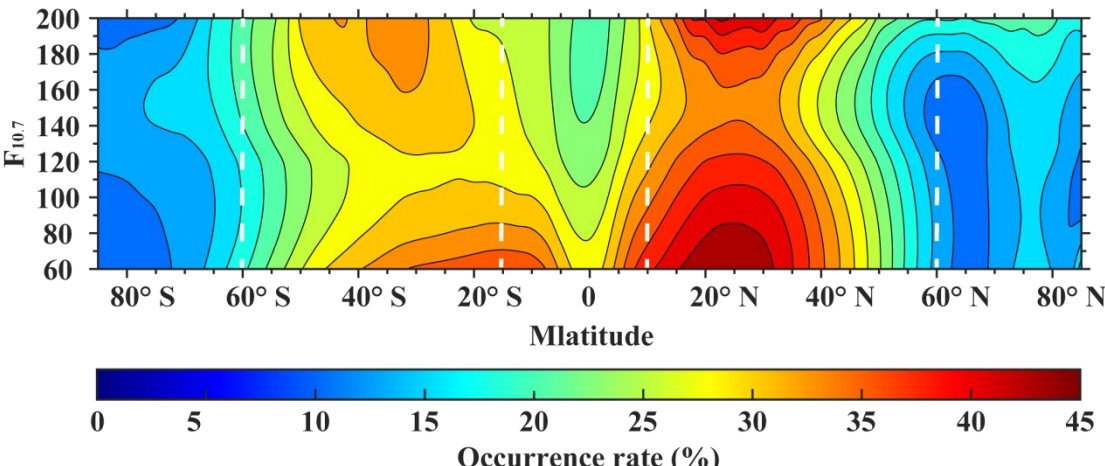

Figure 7: The distribution of Es layer occurrence rate in magnetic latitudinal-$F_{10.7}$ cross-section.

## 4 Discussion

In the above analysis, the response of the Es layer to the geomagnetic and solar activities is investigated. Our results demonstrate that the response of the Es layer to the geomagnetic and solar activities shows latitudinal dependence.

### 4.1 Geomagnetic Variability

As shown in Figures 3 and 4, the relation between the Es layer occurrence and geomagnetic activity presents significant geomagnetic latitudinal dependence. In the high magnetic latitude region from 60° N/S to 80° N/S, the region of the aurora zone, Es layer occurrence rate shows a positive correlation with the geomagnetic and solar activities. High-latitude Es layers are formed, modified, or transported by the action of ionosphere electric fields. It is well known that variations in the spatial distribution of electric fields in the polar ionosphere are driven by the orientation of the interplanetary magnetic field (IMF) embedded within the solar wind. During the geomagnetic disturbed period, the IMF dramatically disturbs, which leads to the disturbance of the ionospheric electric field. Apart from the electric field effect, Kirkwood and Nilsson (2000) proposed that tidal wind shear still plays a significant role in the formation of the high-latitude Es layer. They suggested that the semidiurnal tidal winds dominate control of the daytime Es layers between 100 - 130 km altitude, while the electric fields dominate the formation of nighttime Es layers which are mostly seen below 110 km altitude. Based on the measurements at Chatanika, Alaska (65° N), Sondrestrom, Greenland (67° N), and EISCAT, Scandinavia (69.5° N), Johnson et al. (1987) and Johnson & Virdi (1991) found that the drastic influence of a geomagnetic storm on the zonal wind in the MLT region. MLT neutral wind changes and rapid increase of the tidal amplitude are also observed over Scandinavia during a geomagnetic storm accompanied by solar proton fluxes (Pancheva et al., 2007). Therefore, both the change of ionospheric electric field and wind during the geomagnetic disturbed condition are the possible reasons for the increased occurrence of the Es layer in high latitudes.



In the middle geomagnetic latitudes, the Es layer occurrence rate shows a negative correlation with the geomagnetic activity. The deposition and ablation of meteors are the sources of metallic ions which are the component of the Es layer. This indicates that the variation of the meteor can influence the occurrence and intensity of the Es layer. Based on the long-term observation of meteor radars, Campbell-Brown (2019), and Prikryl (1979, 1983) found that meteor rate shows a negative correlation with the geomagnetic Kp or Cp index. They suggested that the inverse correlation can be ascribed to the solar wind-induced 'geomagnetic' heating of the upper atmosphere, and to the subsequent change in the atmospheric density gradient in the meteor zone. During the geomagnetic disturbed period, the high-latitude energy injection from solar or magnetospheric sources causes a thermal expansion of the neutral atmosphere, which subsequently extends to low altitudes and equatorward. The thermal expansion could cause upwelling and departures from atmospheric molecular diffusive equilibrium, which leads to disturbance in neutral density and temperature in the upper atmosphere, a larger atmospheric scale height, and a smaller neutral density gradient (Buonsanto, 1999; Sutton et al., 2005; Yuan et al., 2015). Recently, Yi et al. (2017; 2018) found that the atmospheric density in the MLT region decreases as geomagnetic activity increases based on meteor radar observations. They also found that the effect of strong geomagnetic storms on the MLT region density can extend to higher middle latitudes (outside the auroral region). The influence of Joule and particle heating can penetrate down to the MLT region heights and extend to middle latitudes during strong geomagnetic activity (e.g., Jiang et al., 2014; Sinnhuber et al., 2012). Meteor rate varies with the neutral density gradient, a larger atmospheric scale height and a smaller neutral density gradient cause the meteor to ablate over a longer distance and therefore result in a smaller metallic ion density from the ablated meteoroids in the MLT region (Lindblad, 1978). Hence, the anti-correlation between meteor rate and geomagnetic activity attributed to the atmospheric density change can be the reason that explains the decreased occurrence rate of the Es layer with the Kp index increase. In addition, it can also explain that the Es layer occurrence rate in low latitudes shows no correlation with the geomagnetic activity since it is difficult for the effect of the heating in high latitudes to extend to the MLT region of the low latitude.

Apart from the meteor inject, the wind plays a dominant role in the Es layer formation at middle and low latitudes. Ma et al. (2001) reported that winds in the MLT region turn from poleward to equatorward and have an eastward enhancement during the geomagnetic storm periods. Goncharenko et al. (2004) found the tidal pattern of neutral winds in the lower thermosphere was heavily disrupted during intense geomagnetic storms (Kp < 8-9). In recent years, based on the TIMEGCM, Li et al. (2019) investigated the wind disturbance in the MLT region during the geomagnetic storm and found that the pressure gradient force associated with vertical wind-induced temperature changes and the Coriolis force are the dominant storm-time momentum forcing processes in the MLT region at middle latitudes during the storm period. Therefore, the wind disturbance during geomagnetic activity can also be the possible reason that explains the relation between the Es layer and Kp index.

**4.2 Solar Variability**

As shown in Figures 6-7, the Es occurrence rate presents a negative correlation with the solar activity in the low and middle geomagnetic latitudes. Based on the meteor radar observations, previous studies investigated the relation between meteor





rates and solar activity (e.g., Campbell-Brown, 2019; Lindblad, 1976; Premkumar et al., 2018). The observational results

demonstrate that there is a negative correlation between meteor rate and solar activity. The strong negative correlation may be attributed to the variation of the density gradient caused by solar activity. During solar minimum, a smaller scale height and steeper density gradient cause meteors to ablate over a shorter distance and reach a higher maximum brightness and therefore a larger plasma line density and more metallic ions. However, during solar maximum, heating in the atmosphere causes scale height to increase and meteors of the same mass are fainter, hence lead to the decrease in plasma line density

and metallic ions, which will eventually result in the reduced occurrence of the Es layer.

Note that, as shown in Figure 7, the Es layer occurrence rate first decreases with the increase of the F10.7 index, however, it shows an increasing trend after the F10.7 index exceeds a certain value (150 in the northern hemisphere and 120 in the southern hemisphere). Maksyutin and Sherstyukov (2005) reported that the low intensive Es layer with critical frequency foEs > 3 MHz and foEs > 4 MHz shows a positive correlation with the solar activity cycle, while the high intensive Es layer

with foEs > 6 MHz and foEs > 7 MHz shows a negative correlation. They attributed the variability of correlation coefficients of the Es layer occurrence rate with the solar activity level to the difference in ion composition of Es layers with different intensities. That is, high-intensive Es layers are formed from metallic ions mainly (Whitehead, 1970; Chavdarov et al., 1975; Whitehead, 1989; Carter & Forbes, 1999), while weak intensive Es layers are formed from molecular ions. Therefore, the increasing trend of Es occurrence rate, as presented in Figure 7, could be related to the low intensive Es layers formed from

molecular ions.

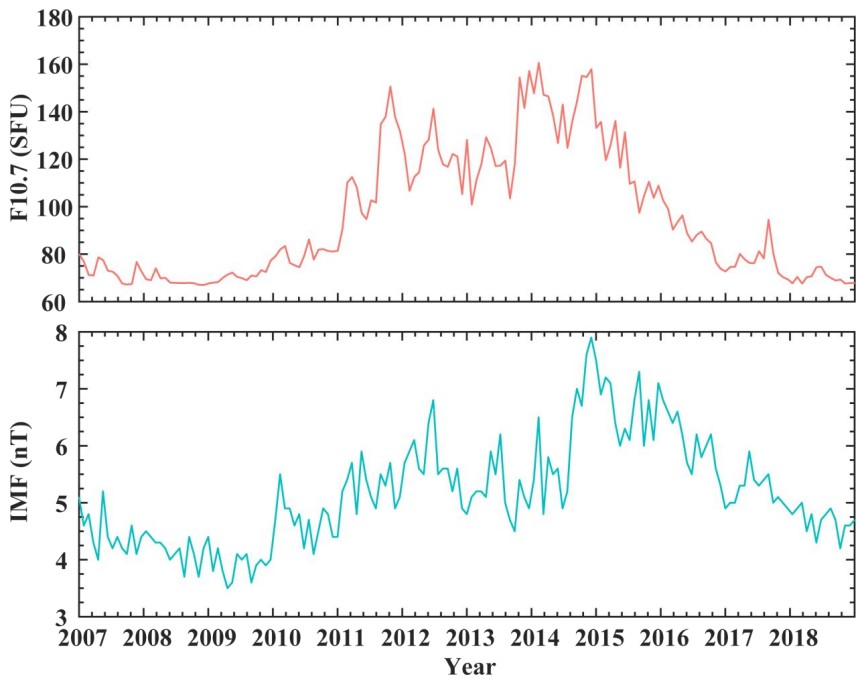

**Figure 8: Variations in the interplanetary magnetic field (IMF) strength and solar flux intensity ($F_{10.7}$) between 2007 and 2018.**



In the high geomagnetic latitudes, there is a positive correlation between the Es layer occurrence rate and solar activity. As mentioned above, the Es layer formation at high latitude is mainly controlled by the ionospheric electric field, while the electric field is driven by the IMF. Numerical studies have shown that the strength of the IMF can vary by up to 50%, peaking at solar maximum (e.g., Hapgood et al., 1991; Papitashvili et al., 2000). Figure 8 presents the variations in the interplanetary magnetic field (IMF) strength and solar flux intensity ($F_{10.7}$) between 2007 and 2018. The IMF strength increases with the enhancement of the solar flux intensity. This indicates that the increase of Es layer occurrence with enhanced solar activity mostly relates to the variation of IMF.

**5 Summary**

In the study, the latitudinal dependence of Es layers on the geomagnetic and solar activity is investigated. The main results of the study are summarized as follows:

The correlation between the Es layer occurrence rate and geomagnetic Kp index shows obvious latitudinal dependence. In low geomagnetic latitudes, the Es layer occurrence rate shows no correlation with geomagnetic activity. In middle geomagnetic latitudes, the Es layer occurrence rate shows a negative correlation with geomagnetic activity, while it shows a positive correlation in high geomagnetic latitudes.

The correlation between the Es layer occurrence rate and solar activity also shows latitudinal dependence. In low and middle geomagnetic latitudes, the Es layer occurrence rate first presents a negative correlation with solar activity and then presents a positive correlation after the solar $F_{10.7}$ index exceeds a certain value. In the high geomagnetic latitudes, the Es layer occurrence rate shows a positive correlation with solar activity.

*Data Availability. Statement* The COSMIC data is available from the COSMIC data analysis and archive center of University Corporation of Atmospheric Research (http://cdaac-www.cosmic.ucar.edu/). The solar and geomagnetic indexes are obtained from the OMNI website (https://omniweb.gsfc.nasa.gov/form/dx1.html).

*Author contributions*. Qiong Tang and Chen Zhou designed the study and wrote the manuscript. Huixin Liu, Yi Liu  Jiaqi Zhao, and Zhibin Yu contributed significantly to the early version of the manuscript. Yuzheng Zhao and Xueshang Feng contributed to the discussion of the results and the preparation of the manuscript. All authors discussed the results and commented on the manuscript at all stages.

*Competing interests*. The authors declare that they have no conflict  of interest.

*Acknowledgments*. We acknowledge the COSMIC data analysis and archive center of University Corporation of Atmospheric Research (http://cdaac-www.cosmic.ucar.edu/) for the provision of RO scintillation data. This work was



supported by the National Natural Science Foundation of China (NSFC grant No. 41574146, 41774162, 42004130), and the Foundation of the National Key Laboratory of Electromagnetic Environment (Grant No.6142403180204). H. L. acknowledges supports by JSPS KAKENHI (Grants nos. 18H01270 and 17KK0095) and JRPs- LEAD with DFG.

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
