# Peer review of "Latitudinal Dependence of the Geomagnetic and Solar Activity Effect on Sporadic-E layer"

_Atmospheric Chemistry and Physics, 2022_

## Referee Comment (RC3)

Referee report on the paper: "Latitudinal dependence of the geomagnetic and solar activity effect on sporadic E layer", by Tang et al., submitted for publication in ACP-EGU.

One should keep in mind that Es layers are metal ion layers. Es formation and intensity depend primarily on: (a) the metal ion abundances in the E region and their dynamic changes and spatial distribution, (b) the tidal wind system that provides the vertical wind shears needed for layer forming and built up, and (c) the horizontal component of the Earth's magnetic field. These are key parameters which determine the diurnal, seasonal, altitudinal, latitudinal, and longitudinal Es variations. On the other hand, there are other minor causes of Es variability as well, among them possibly those related to geomagnetic and/or solar activity.

This paper searches for effects of geomagnetic (Kp) and solar activity (F10.7) on sporadic E (Es) layer occurrence in different geomagnetic latitude zones (low, middle, and high latitudes). It uses COSMIC radio occultation observations over a period of a solar cycle, 2007 to 2018.

The topic dealt with in this work has been investigated by several others in the past, using mostly ionosonde observations (see review paper by Whitehead, 1989). These studies showed either there are no geomagnetic and solar activity influences on Es or that such effects are minimal compared to variations caused by other sources. I value a study by Pietrella and Bianchi (2009), which relied on 32 years of continuous measurements obtained with the Rome ionosonde, which showed no geomagnetic or solar cycle effects on midlatitude Es. In addition, and in relation with our understanding of sporadic E, there is no solid physical base as to why Es should depend on solar and geomagnetic variations. An experimental study of geomagnetic and/or solar effects on Es requires careful consideration and analysis.

Saying all this, following is a list of shortcomings that cast doubt on the importance of this work.

1). Although the authors know what the key causes of Es variability are, they make no effort to reduce their effect on the statistics of what they are searching for, this is, of possible influences caused by geomagnetic and solar cycle variations. In the present statistical treatment, "everything is thrown in the pot" hoping that in the end it will survive only the effects of what the authors look for. This is a simplistic way to go. For example, one would improve his statistics by selecting to look at a fixed location, a fixed longitudinal sector, or a given season to perform his analysis; this is not done here. In addition, the authors ignore in their geomagnetic variability part of their study that the statistics there can be affected by the solar cycle activity as well, knowing that geomagnetic and solar cycle variations are closely related (see their Figure 8).

2). Nothing is said about cleaning the data from unreliable S4 data samples. Regarding this, see a recently published paper (not cited here) by Bergsson, B., & Syndergaard, S. (2022). Global temporal and spatial variations of ionospheric sporadic-E derived from radio occultation measurements. Journal of Geophysical Research: Space Physics, 127, e2022JA030296. https://doi. org/10.1029/2022JA030296. These authors use the same COSMIC RO data set as in the present study by Tang et al. and apply a new method of analysis that excludes erratic measurements.

3). At high (auroral) latitudes, the RO-based Es signatures can be contaminated with, or be mistaken by, signal variations caused by field-aligned irregularities in relation with instabilities in the auroral electrojets (E region ExB currents). These horizontal currents, known as eastward and westward auroral electrojets, are set up mostly at nighttime by enhanced electric fields and energetic particle precipitation, both of magnetospheric origin. These drivers intensify during geomagnetically disturbed conditions and cause plasma turbulence through instabilities that produce intense field-aligned irregularities (FAI) in the altitude range between 95 and 115 km. These auroral E region irregularities are expected to contaminate the RO sporadic E layer events used in the present analysis. The authors do not mention this problem, which could become important for their high latitude data during disturbed geomagnetic conditions.

4). I find the use of the circular-type color plots in Figure 1,2, and 5 terribly awkward and difficult to inspect. The proper thing to do here is to use orthogonal axes for time (x-axis) and altitude (y-axis), like everybody else does. This helps the reader to interpret better the information and compare them with other studies.

5) Since the authors do comparisons and search for latitudinal differences in their plots, they should have kept the percent occurrence scaling the same everywhere and not change it around. Also, they do not provide the number of RO Es occurrence samples corresponding to each plot, which is an indication of statistical stability (see next point). Furthermore, it is important to have included deviations from the mean for the occurrence histogram bars in Figures 3 and 6.

6). The number of data samples used here for different latitudes and geomagnetic and solar activity levels are highly uneven, which affects the comparison of the results. This means, one cannot confidently compare effects in middle latitudes where Es occurrence is overwhelmingly frequent, with those at high latitude where Es layers are sparse leading to a small number of RO samples (see several published global maps of RO-based Es occurrence measurements). The same is true for low and high Kp conditions, since high Kp values are much less common than lower. Simply, limited data sets make statistics unstable and inconclusive. The latter is clearly noted in Figure 2 for the high Kp plots, which can hardly be used to arrive at definitive conclusions on geomagnetic dependences during times of low and high Kp conditions.

7). Following here is an important point: Careful inspection of all Figures in the paper shows that the authors' description and interpretation of the statistics is strongly subjective, which makes their findings questionable. I will comment briefly on the most comprehensive plots of the paper: the occurrence distributions for different Kp ranges in Figure 3, and those for different A10.7 index ranges in Figure 6, which refer to geomagnetic and solar variations, respectively.

   (a) Based on Figure 3, the authors state that "Es correlation with Kp is positive at high latitudes". I note that this is true only for latitudes between 60 and 70 deg, which are largely auroral zone geomagnetic latitudes. However, at these latitudes the systematic positive correlation could be related with field-aligned electrojet irregularities that become stronger with higher geomagnetic activity. The fact that at these latitudes the data samples occur during nighttime, points to electrojet FAI rather than to Es layer occurrences. This option is ignored in the present analysis. On the other hand, for latitudes greater than 70 deg the trends are small and

of variable polarity, thus there is no convincing evidence here that Es occurrence systematically increases with Kp at (all) high latitudes.

(b) It is stated that "correlation with Kp is negative in middle geomagnetic latitudes". Careful inspection of the histogram plots in Figure 2 shows that this conclusion does not apply for all the midlatitude cases shown there and different Kp ranges. Also, the trends between the north and south hemisphere differ, which is not expected if these were supposed to be caused by global scale geomagnetic effects.

(c) Regarding now the solar activity effects on Es layers that occur during small Kp conditions (see Figure 6), the authors conclude that there is "a latitudinal dependence with negative correlation in low and middle latitudes and positive correlation in high geomagnetic latitudes". Inspection of Figure 6 shows that this is an overstatement. Simply, the differences in Es occurrence with solar activity are minimal from plot to plot and thus one cannot rely on these statistics to conclude that there is a clear correlation between sporadic E and solar activity. To put it in a few words, the evidence is not sufficient to support this conclusion.

8). In their discussion section, the authors try to explain their findings by referring selectively to a few publications. This is done in a general, very speculative, and biased way without providing details and convincing arguments. The discussion in section 4.2 (solar variation) is not necessary because, as mentioned briefly above in 7c, the evidence shown in section 3.3 is not conclusive/ definitive. The physical explanations summarized in the abstract cannot be supported by the present statistical findings.

9). One would have expected the authors to discuss their findings with published results on the same topic, but this is not done. Especially, it is striking that there is no discussion with regards a relatively recent paper (2017) authored by several of the same authors as in the present paper: C. Zhou, Q. Tang, X. Song, H. Qing, Y. Liu, X. Wang, X. Gu, B. Ni, and Z. Zhao (2017), A statistical analysis of sporadic E layer occurrence in the midlatitude China region, J. Geophys. Res. Space Physics, 122, 3617-3631, doi:10.1002/ 2016JA023135). In that paper, the authors used a latitudinal chain of ionosondes in China to report that: "*The occurrence of the midlatitude sporadic E layer tends to increase with the level of geomagnetic activity on the basis of both the statistical analysis and a case study*". The authors say nothing about this earlier finding (of their own) that contradicts with that reported in the present study, this is: "*The Es layer occurrence decreases with Kp at midlatitude*".

10). There are several unclear and some misleading and wrong statements or points made throughout the text. The overall presentation is far from optimal.

*Conclusion.* Based on the serious shortcomings of the present work, which are explained and argued above, I consider the statistical evidence included in the paper to be insufficient to substantiate the conclusions made by the authors. Therefore, I cannot recommend the paper's publication in "Atmospheric Chemistry and Physics" and urge the authors to not resubmit this paper in some other journal.